# Droplet Generation in a Flow-Focusing Microfluidic Device with External Mechanical Vibration

**DOI:** 10.3390/mi11080743

**Published:** 2020-07-30

**Authors:** Zhaoqin Yin, Zemin Huang, Xiaohui Lin, Xiaoyan Gao, Fubing Bao

**Affiliations:** Institute of Fluid Measurement and Simulation, China Jiliang University, Hangzhou 310018, China; yinzq@cjlu.edu.cn (Z.Y.); huangunderstand@163.com (Z.H.); hebut_ck12_lxh@163.com (X.L.); gao_star@163.com (X.G.)

**Keywords:** droplet generation, external mechanical vibration, flow-focusing microfluidic device

## Abstract

The demand for highly controllable droplet generation methods is very urgent in the medical, materials, and food industries. The droplet generation in a flow-focusing microfluidic device with external mechanical vibration, as a controllable droplet generation method, is experimentally studied. The effects of vibration frequency and acceleration amplitude on the droplet generation are characterized. The linear correlation between the droplet generation frequency and the external vibration frequency and the critical vibration amplitude corresponding to the imposing vibration frequency are observed. The droplet generation frequency with external mechanical vibration is affected by the natural generation frequency, vibration frequency, and vibration amplitude. The droplet generation frequency in a certain microfluidic device with external vibration is able to vary from the natural generation frequency to the imposed vibration frequency at different vibration conditions. The evolution of dispersed phase thread with vibration is remarkably different with the process without vibration. Distinct stages of expansion, shrinkage, and collapse are observed in the droplet formation with vibration, and the occurrence number of expansion–shrinkage process is relevant with the linear correlation coefficient.

## 1. Introduction

The technology of dealing with fluids on a microscale have made microfluidics an important tool for many biological and chemical applications [1,2,3]. Widespread interest in the droplet-based microfluidics for drug transportation [4], cell cultivation [5,6], material synthesis [7,8,9], etc. has emerged due to the attractive advantages of the small amount of sample needed, high surface/volume ratio, massive throughput of droplets, effective control of each droplet, and so on [1,10,11,12]. Therefore, the droplet generation system should be able to produce the stable droplet stream in a reproducible way, as well as have the capacity of controlling the formation and transportation of droplet in a prescribed manner [13].

In general, droplets can be produced in microfluidic devices using two methods: the passive way and the active way [10,14]. In the passive way, the dispersed phase is introduced into the continuous phase to form the droplets, which is the result of a competition of viscous, inertial and surface tension forces. Jeong et al. [15] employed a 3D flow-focusing microfluidic device to improve the stability of the submicron emulsion droplet generation, and the emulsion droplet could be controlled by the pressure ratio. Jakiela et al. [12] investigated the parameters affecting the speed of droplet flow and revealed that the droplet speed significantly relies on the length of droplet, the capillary number and the viscosity ratio between the two liquids. While in the active way, additional energy (such as electric, magnetic, acoustic, etc.) provided by the external elements is applied for controlling the droplet formation [10]. Link et al. [16] presented a platform technology based on the electrostatic charge of droplet and the external electric field to control the droplet generation. Both the direct current field [17] and the alternating current field [18] can be applied on microfluidic devices to generate the droplets. Tan et al. [19] utilized a circular permanent magnet to manipulate the ferrofluid droplet formation in a T-junction microfluidic device, and reported that the additional magnetic force could cause the variation of droplet size. Park et al. [20] employed pulse laser on the microfluidic platform to produce droplet, and the droplet generation frequency could reach 10,000 Hz in their experiments, meanwhile the droplet volume could be tuned continuously in the range of 1–150 pL. Moreover, the aid of surface acoustic waves for manipulating the fluids in microfluidic systems has gained more and more attraction [3,21]. Cheung and Qiu [22,23] presented the evolution of interfacial profile during the droplet generation with the surface acoustic actuation, and investigated the effects of acoustic voltage, acoustic frequency, fluid viscosity on the droplet size. Collins [24] used surface acoustic waves to produce droplets on picoscale, and the droplet volume is controlled by the applied power, the acoustic force duration and the channel geometry. Although the active methods involving electric, magnetic, laser, acoustic elements have the capacities of controlling the droplet formation and manipulating droplet transportation in a fast-responding and effective way [10], the requirement of new-sets of on-chip integration components dramatically raises the price of microfluidic device fabrication [11,25].

Therefore, the microfluidic device coupled with external mechanical vibration is of great significance in the active droplet generation, because only an off-chip vibration excitation device is required [11]. Comparing with on-chip devices (such as piezoelectric control, microvalve, etc.), the off-chip mechanical vibrator is simpler in assemble and easier compatible with microfluidic device [10]. Sauret and Shum [26] used a mechanical vibrator to impose the direct pressure perturbation on the dispersed phase in a co-flowing microfluidic device, which allowed the direct generation of the single emulsions in water-in-water flow as well as the double emulsions in water-in-water-in-water flow. They pointed out that the incorporation of mechanical vibrator is the easier way to generate droplet in the aqueous two-phase systems. Later, Sauret et al. [27] further imposed harmonic perturbations on the dispersed phase of the oil-water system, and the results showed that the implement of perturbation could cause the formation of liquid jet with unique structures affected by the fluid flow rates and perturbation. Zhu et al. [11] also introduced the mechanical vibration in a co-flow microfluidic device to investigate the formation frequency and uniformity of the dispersed droplets. The previous works have shown that the microfluidics coupling external mechanical vibration could be a promising method in active droplet generation, since this method is easy to implement, and able to precisely control and manipulate the droplets [27,28]. Meanwhile, the mechanical vibrator can be applied to all-aqueous systems even in the systems with ultra-low interfacial tension (such as the oil-in-water flow with great amounts of surfactants) [27]. However, the mechanisms of the active droplet generation with mechanical vibration in microfluidic devices and the relation between the droplet generation frequency and the external vibration condition have not been comprehensively studied in the existing references.

In the present study, therefore, the droplet generation with external mechanical vibration is experimentally studied in a flow-focusing microfluidic device. The effects of vibration frequency and vibration amplitude on the droplet generation were investigated, and the relationship between the droplet generation frequency and the imposing vibration frequency was obtained. Moreover, the dynamics of droplet generation with mechanical vibration were explored.

## 2. Experimental Setup

The experimental setup for droplet generation in a flow-focusing microfluidic device with external mechanical vibration is shown in Figure 1. The experiment system consists of a basic droplet generation system and an external mechanical vibration system. The basic droplet generation system is mainly composed of a polydimethylsiloxane (PDMS) based flow-focusing microfluidic chip, two syringe pumps (Harvard Pump 11 Elite, Hollision, MA, USA), an inverted microscope (Nikon Eclipse Ti-S, Nikon, Tokyo, Japan), a high-speed camera (NAC MEMRECAM HX-6, Vehicle Test System Ltd., Shanghai, China), etc. Figure 2 shows the geometry of the flow-focusing channels. This microfluidic device is composed of a dispersed phase inlet channel (inlet 1), two continuous phase inlet channels (inlet 2), and an outlet channel, respectively. The widths of inlet channels (*w_d_*, *w_c_*) and the outlet channel (*w_o_*) are all 100 μm. The width of the aperture (*w_m_*) connecting to outlet channel is 50 μm. The length of the connecting aperture (*w_n_*) is 25 μm. And the heights of all channels are 38 μm.

In addition, the external vibration system is mainly composed of a signal generator (RIGOL DG4062, Rigol, Beijing, China), a mechanical vibration exciter (JZK-10A, Sinocera Piezotronics.INC, Jiangsu, China), a power amplifier (YE5872A, Sinocera Piezotronics.INC, Jiangsu, China), and an acceleration sensor (PCB PIEZOTRONICS 301A11, Ti, Dallas, TX, USA). The sinusoidal waves are produced by the signal generator and amplified by the power amplifier; and then the amplified signals cause the corresponding vibration of the mechanical vibration exciter. In our experiments, the mechanical vibration exciter is applied on the polytetrafluoroethylene (PTFE) microtube connecting to inlet of the dispersed phase (i.e., inlet 1 in Figure 2) with the distance of 27 cm between the mechanical vibrator and the inlet 1. Thus the microtube vibrates in the gravitational direction consistently with the external vibration exciter, resulting in the periodic pressure variation in the dispersed phase channel [11]. The sinusoidal displacement of the imposed vibration can be expressed by:(1)$$y\left(t\right)=\varepsilon \sin\left(2\pi f_{P}t\right)$$
where *y*(*t*) is the vibration displacement at t moment; *f_P_* is the frequency of the vibration; and ε is the amplitude of the vibration. In our experiments, the maximum acceleration (*a_m_*) of sinusoidal vibrations, which is measured by the acceleration sensor assembled on the connecting shaft of the vibration exciter is measured and discussed. In addition, the frequency and amplitude of the external vibration can be controlled by the signal generator and power amplifier.

A mixture of 4 wt.% surface-active agent (EM-90, Degussa, SigmaAldrich, Saint Louis, MO, USA) and 96 wt.% mineral oil (Sigma-M5904, SigmaAldrich, Saint Louis, MO, USA) is chosen as the continuous phase, and a mixture of deionized water and a small amount of polyvinyl alcohol is applied as the dispersed phase. The viscosity of the continuous phase (*μ_c_*) is 23.8 mPa⋅s, and the viscosity of the dispersed phase (*μ_d_*) is 0.92 mPa⋅s, which are measured by a viscometer (Lichen NDJ-5S, LICHEN, Shanghai, China). The surface tension (*γ*) is 8.3 mN/m, which is measured by a Wilhelmy plate tensiometer (Shanghai BZY-2, HengPing Instrument and Meter Factory, Shanghai, China). The high-speed camera runs at a frame rate of 8000 fps to capture the droplet formation in flow-focusing devices. The droplet generations under each condition are performed at least 3 times to ensure the reliability, and all experiments are carried out at a room temperature of 25 ± 0.1 °C.

## 3. Results and Discussions

### 3.1. Effect of Vibration Frequency on Droplet Generation Frequency

To investigate the droplet generation process with external mechanical vibration, the effects of vibration frequency (*f_P_*) and vibration amplitude (*a_m_*) on the generation frequency (*f_G_*) are first studied. The effect of vibration frequency is studied by varying the frequency of external mechanical vibration in the range of 0–500 Hz while the maximum acceleration is held at 60 m/s^2^. To study the effect of vibration amplitude, the external vibration frequency is kept constant at a certain frequency and the vibration amplitude varies through the control of acceleration.

When the flow rates of continuous phase (*Q_c_*) and dispersed phase (*Q_d_*) are 90 and 60 μL/h, respectively, the droplet generation frequencies at different vibration frequencies are shown in Figure 3. The droplet generation frequency without external vibration (0 Hz), named as the natural generation frequency (*f_N_*), is 44 Hz. As shown in the figure, it is obvious that the vibration in the dispersed phase plays an important role on the droplet generation frequency. When the external vibration frequency increases from 40 to 500 Hz, the generation frequency of discrete droplet fluctuates in the range of 40–170 Hz.

Promoting the vibration frequency from 40 to 80 Hz, the droplet generation frequency is consistent with the external vibration frequency. However, corresponding to the vibration frequencies of 90 and 100 Hz, the generation frequencies of droplet are 45 and 50 Hz, respectively, which are only half of the vibration frequencies applied. Continuously increasing the vibration frequency to the range of 110–500 Hz, it is interesting that the droplet generation frequency becomes consistent with the vibration frequency again in the range of 110–130 Hz, whereas the droplet generation frequency decreases sharply to 53.5 Hz as the vibration frequency reaches to 160 Hz. Then the generation frequency soars to 170 Hz which is in accordance with the external vibration frequency, thereafter, the generation frequency of dispersed droplet maintains around 50 Hz, as the vibration frequency varies in the range of 200–500 Hz.

The relation between the droplet generation frequency and the external vibration frequency is given in Figure 4. As shown in the figure, linear correlation can be obviously observed, thus the droplet generation frequency can be expressed as follow:(2)$$f_{G}=k\cdot f_{P},\;\left(k=1,\;1/2,\;1/3,\;1/4,\;1/5\right)$$
where *k* = 1 means the generation frequency of dispersed droplet is synchronized with the external mechanical vibration frequency, that is one external vibration can produce one droplet (such as the vibration frequency of 40 Hz). While *k* = 1/2 indicates the droplet generation frequency is only half of the external vibration frequency, in other words, it needs two external vibrations in the dispersed flow to generate one droplet (such as vibration frequency of 90 Hz). Furthermore, relationships of *k* = 1/3, ¼, and 1/5 are also observed in our experiments, as shown in Figure 3 and Figure 4.

Although the linear correlation between droplet generation frequency and vibration frequency hasn’t been mentioned in previous studies on the active droplet generation with mechanical vibration, it is found that Equation (2) is also applicable to the experimental results carried out by Zhu et al. [11] on the droplet generation with external mechanical vibration in a co-flowing microfluidic device. The relevant values of droplet generation frequency and mechanical vibration frequency in reference [11] are cited and shown in Figure 5. Apparently, the relations of *k* = 1, 1/3 and 1/4 can be observed. However, the relations of *k* = 1/2 and 1/5 are not found, this may be related to the differences in device geometry and vibration condition.

### 3.2. Effect of Vibration Amplitude on Droplet Generation Frequency

Figure 6 shows the variation of droplet generation frequency with the increasing vibration amplitude when the vibration frequency is held at 170 Hz. The flow rates of continuous phase and dispersed phase are both 60 μL/h, correspondingly, and the natural generation frequency of dispersed droplet is 32 Hz. It is clear that the droplet generation in the flow-focusing device is greatly affected by the increase of vibration amplitude, as the external vibration amplitude increases from 0 m/s^2^ (without vibration) to 136.3 m/s^2^, the generation frequency of dispersed droplet increases monotonously in the range of 32–170 Hz, and the volume of generated droplet decreases correspondingly. When imposing external vibration on the dispersed phase and increasing the vibration amplitude, the droplet generation frequency maintains at the natural generation frequency of 32 Hz at first, till the vibration amplitude reaches to 10.6 m/s^2^, the frequency of generated droplet jumps to 34 Hz which is one fifth of the vibration frequency (170 Hz). Continuously promoting vibration amplitude, the droplet generation frequency keeps at 34 Hz before the amplitude reaches to 19.1 m/s^2^, and then the generation frequency skips to one quarter of the vibration frequency (43 Hz). Similarly, after held at 43 Hz, the droplet generation frequency arrives one third of the vibration frequency (58 Hz), corresponding to the vibration amplitude of 30.6 m/s^2^. When the vibration amplitude gets to 39.9 m/s^2^, the generation frequency jumps to 85 Hz. Finally, the droplet generation frequency reaches to 170 Hz, synchronizing with external vibration, as the vibration amplitude is greater than or equal to 136.3 m/s^2^. Obviously, the relation between the droplet generation frequency and external vibration frequency under different vibration amplitudes also can be expressed by Equation (2). In other words, the step changes happen in droplet generation frequency with the increases of vibration amplitude in the range of 0–136.3 m/s^2^.

The relation between the droplet generation frequency and the external vibration amplitude is shown in Figure 7. It can be observed that the droplet generation frequency generally increases with the rising vibration amplitude when the external vibration frequency is constant. Moreover, there is a critical vibration amplitude corresponding to the imposing vibration frequency, and the droplet generation frequency will be synchronized with the vibration frequency if the vibration amplitude is equal to or greater than the critical amplitude. It also can be indicated in Figure 7 that the minimal droplet generation frequency obtained with the imposing vibration amplitude less than the critical vibration amplitude is natural generation frequency.

In addition, the critical amplitudes at various vibration frequency are measured under the flow conditions of *Q_c_* = 90 μL/h and *Q_d_* = 60 μL/h (as shown in Figure 8). No obvious functional relationship between the critical amplitude and the imposing vibration frequency could be deduced from the experimental data, since the critical vibration amplitude fluctuates with the increasing vibration frequency. However, it can be found in Figure 8 that the critical amplitudes corresponding to the vibration frequencies in the ranges of 85–105 Hz and 135–160 Hz are greater than 60 m/s^2^. Therefore, this is the reason why the droplet generation frequency is lower than the vibration frequency in some cases when the vibration amplitude is held at 60 m/s^2^ (as presented in Section 3.1 above).

Through the combination of effects of vibration frequency and vibration amplitude on droplet generation frequency, it can be indicated that the droplet generation frequency with external mechanical vibration is affected by the natural generation frequency, vibration frequency, and vibration amplitude. Since the natural generation frequency is determined by the flow condition in a certain microfluidic device, the droplet generation frequency with external vibration will vary from the natural generation frequency to the imposed vibration frequency at different vibration conditions.

### 3.3. Dynamics of Droplet Generation

Two typical processes of droplet generation in the flow-focusing device are illustrated in Figure 9, with and without the external vibration. The flow rates of continuous phase and dispersed phase are both 60 μL/h, the natural droplet generation frequency at this flow condition is 32 Hz (see Figure 9a), while the external vibration with the frequency of 170 Hz and the acceleration amplitude of 40 m/s^2^ is applied on the dispersed phase, the droplet generation frequency at the same flow condition is 170 Hz.

Without vibration: as the dispersed phase thread breaks to form a droplet, a new generation cycle starts (defined as 0 ms). At the beginning, the fluid thread of dispersed phase retracts in the axial direction while expands in the radial direction due to the effects of surface tension and dispersed phase (around 0–4 ms). Then, the dispersed phase thread grows in both axial and radial directions, and the thread tip is shaped like a bullet (see Figure 9a, 8.0 and 12.0 ms). As the expansion continues, the dispersed phase thread reaches the aperture and plugs into the outlet channel (see Figure 9a, 16.0 ms). Later, the radial thread of dispersed phase is thinned by the continuous phase, thus a neck appears in the intersection region and the dispersed thread begins to collapse. Since the dispersed phase thread grows and moves downstream continuously, the thread in the outlet channel expands progressively to block the channel, however, the thread neck thins gradually. Lastly, the thread neck is pinched off, and a dispersed droplet is generated. The droplet generation without vibration in our study is similar to the droplet formation in the dripping regime reported by Fu et al. [29], as the droplet generation processes go through three distinct stages: retraction, expansion, and collapse. However, this is quite different with the experimental work presented by Wu et al. [30] where no retraction was observed in the ferrofluid droplet formation process in the flow-focusing device. Since the retraction of dispersed phase is resulted from the surface tension [30], the much larger surface tension in our experiments could be the main reason cause the differences.

With vibration: the pinch-off moment of the dispersed phase thread is also defined as the beginning of a new droplet formation cycle. It can be obviously observed that the droplet generation process with external vibration is of great differences with the process without vibration. As the dispersed phase thread breaks in the vicinity of the aperture, firstly, it expands in both axial and radial directions and plugs into the outlet channel. The dispersed thread grows rapidly in the intersection and outlet channel, and its volume reaches the maximum at 2.3 ms. As a result, the last droplet deforms heavily when moving downstream. Secondly, the dispersed phase thread shrinks and the previously generated droplets move upstream, which are caused by the pressure drop in the dispersed phase. Thirdly, the thread thins in the intersection, thus a neck is present and shrinks gradually. Meanwhile, the thread volume in the outlet channel decreases and reaches the minimum at 5.3 ms, and then, the dispersed phase thread turns to expand again while the thread neck thins continuously. Finally, the dispersed phase thread is pinched off to generate a new droplet. Figure 9b shows an example of droplet generation with vibration when the droplet generation frequency is synchronized with vibration frequency. Apparently, the process of droplet generation with vibration in this case can be divided into three stages: expansion, shrinkage, and collapse. The pressure fluctuations in the dispersed phase caused by the external mechanical vibration greatly contribute to the expansion and shrinkage of dispersed phase thread, affecting the droplet generation process significantly.

When the vibration is imposed on the dispersed phase, no retraction happens, however distinct expansion and shrinkage are observed which represent the main differences resulting from droplet generation without vibration. The volume variations of thread head during droplet generation process introduced above are presented in Figure 10. The dispersed phase thread between the edge of the inlet 1 channel and the thread tip is defined as the thread head, and its volume (*V_t_*) is obtained by using image J software. Without vibration, the head volume of dispersed phase (*V_t_*) increases monotonously during the whole droplet generation cycle (Figure 10a), since the dispersed phase is continuously pushed into the channel by the syringe pump. While the droplet generation frequency is synchronized with vibration frequency (170 Hz, in Figure 10b), the *V_t_* first increases quickly and reaches the maximum volume at 2.3 ms, then decreases to a minimum volume at 5.0 ms, later the *V_t_* increases again, finally the dispersed phase thread breaks at 5.9 ms to form a new droplet. The processes of expansion, shrinkage and re-expansion are evidently revealed in Figure 10b.

As discussed in Section 3.1 and Section 3.2, the droplet generation frequency can be controlled by the external vibration. Thus, the processes of droplet generation under different vibration conditions are also investigated, and Figure 11 shows the volume variations of thread head at different vibration amplitudes (the vibration conditions are consistent with the experimental conditions in Figure 6). Because the relationship between the droplet generation frequency and the external vibration frequency can be expressed by Equation (2), the corresponding values of coefficient *k* for Figure 11 are 1/5 (34 Hz), 1/4 (43 Hz), 1/3 (57 Hz), 1/2 (85 Hz), and 1 (170 Hz), respectively. It’s interesting that the head volume (*V_t_*) presents diverse peaks during droplet generation process once the external vibration is employed, and the number of peaks is equal to the reciprocal of coefficient *k*. Moreover, one peak of the head volume indicates the expansion–shrinkage process of the dispersed phase thread. For example, when the droplet generation frequency is 34 Hz (*k* = 1/5), the head volume of dispersed phase thread presents five peaks during the whole droplet generation cycle, and the expansion–shrinkage process appears five times before the breakup of the dispersed phase thread. That is to say, it needs five times of the external vibration to produce a new droplet. In addition to the droplet generation frequency increases with the rising vibration amplitude, when the vibration frequency is kept constant (Figure 6), it can be revealed that peak value of *V_t_* in each expansion–shrinkage process also increases with the rising amplitude (Figure 11), due to the increase of vibrational energy at the larger amplitude. While the *V_t_* at breakup moment decreases with the rising amplitude, which is obviously resulted from the increasing droplet generation frequency at the fixed flow condition.

## 4. Conclusions

The droplet generation mechanism in a flow-focusing microfluidic chip with external mechanical vibration on dispersed phase was studied. The influences of both vibration frequency and vibration amplitude on the droplet generation process were experimentally investigated. The relationship between droplet generation frequency and external mechanical vibration frequency can be expressed the linear equation, which is suitable for droplet generation in both flow-focusing and co-flowing devices. The droplet generation frequency with external mechanical vibration is affected by the natural generation frequency, vibration frequency, and vibration amplitude. The critical vibration amplitude corresponding to the imposing vibration frequency was observed. For a given dispersed and continuous phase flow rates, when the acceleration of external vibration on dispersed phase is small, the perturbation on dispersed phase is small and the droplet generation process is unaffected. With the increase of external vibration strength, the droplet generation frequency was dominated by the external vibration frequency, from 1/5 of the vibration frequency, to 1/4, 1/3, and 1/2, and when the vibration acceleration is large enough, the droplet generation frequency was consistent with the external vibration frequency. When the vibrator was more powerful, 1/6 or even smaller of the external vibration frequency droplet generation frequency can be observed. The droplet generation frequency can be precisely controlled by the external mechanical vibration. The droplet generation process can be precisely controlled by adjusting the frequency and amplitude of the mechanical vibration. Compared with the process without external mechanical vibration, the droplet generation with mechanical vibration typically experiences expansion, shrinkage, and collapse stages, and the occurrence number of expansion–shrinkage process is equal to the coefficient in linear equation. The active droplet generation coupling mechanical vibration could be a promising method in microfluidic applications, since it has the ability to produce droplets in a reproducible and prescribed manner.

## Figures and Tables

**Figure 1 micromachines-11-00743-f001:**
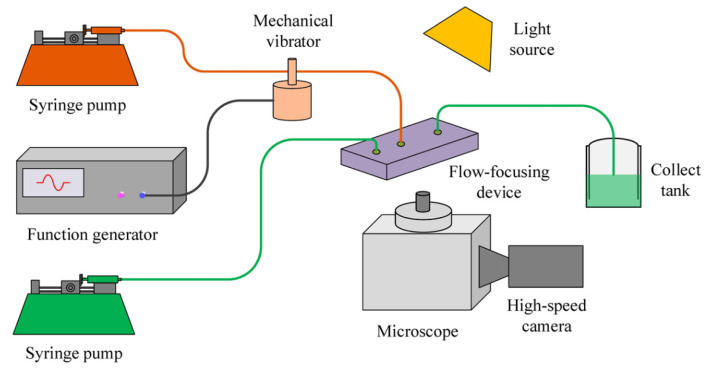
Schematic diagram of experimental setup for droplet generation in flow-focusing microfluidic chip with external mechanical vibration.

**Figure 2 micromachines-11-00743-f002:**
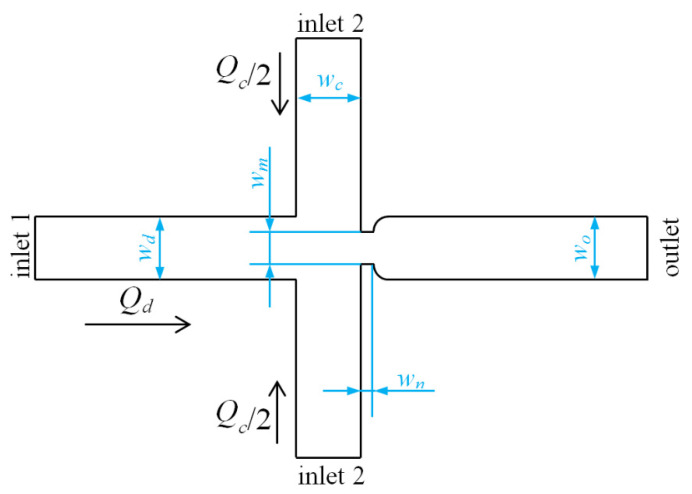
Schematic diagram of the flow-focusing microfluidic chip.

**Figure 3 micromachines-11-00743-f003:**
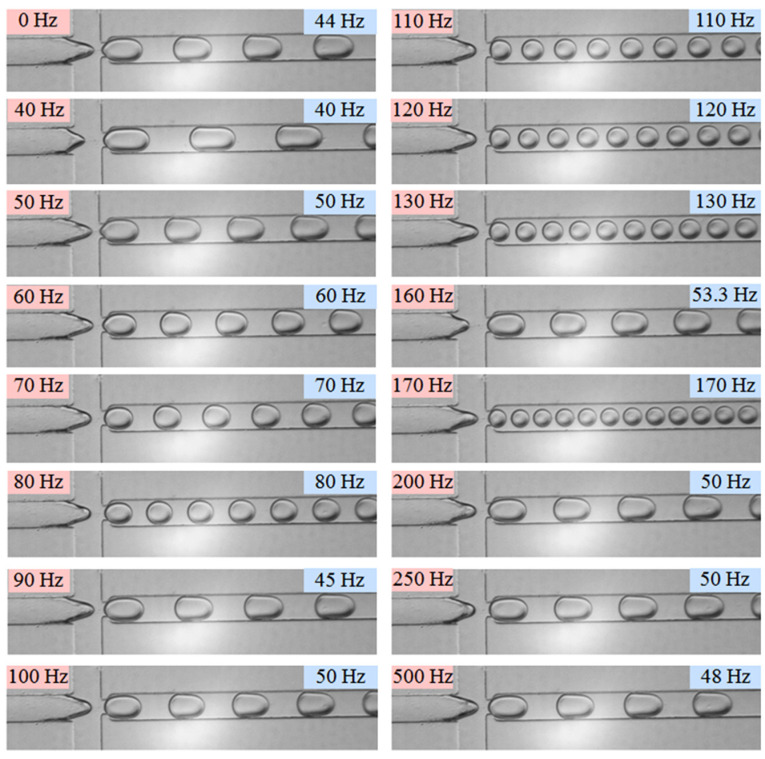
Droplet generation snapshots at different external vibration frequencies. (*Q_c_* = 90 μL/h, *Q_d_* = 60 μL/h, *a_m_* = 60 m/s^2^; *f_P_* are marked on the upper left corner; *f_G_* are marked on the upper right corner.).

**Figure 4 micromachines-11-00743-f004:**
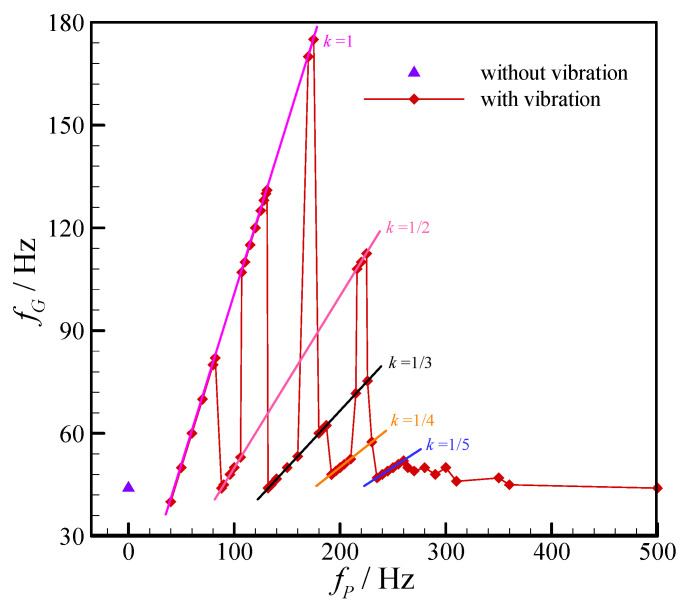
Variation of droplet generation frequency with external vibration frequency in the flow-focusing microfluidic device. (*Q_c_* = 90 μL/h, *Q_d_* = 60 μL/h, and *a_m_* = 60 m/s^2^.).

**Figure 5 micromachines-11-00743-f005:**
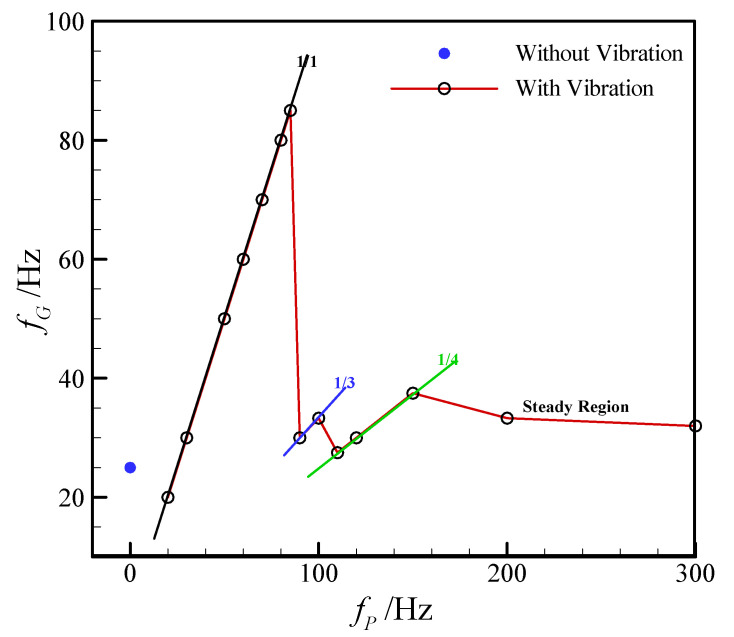
Variation of droplet generation frequency with external vibration frequency in the co-flowing microfluidic device, data are cited from reference [11].

**Figure 6 micromachines-11-00743-f006:**
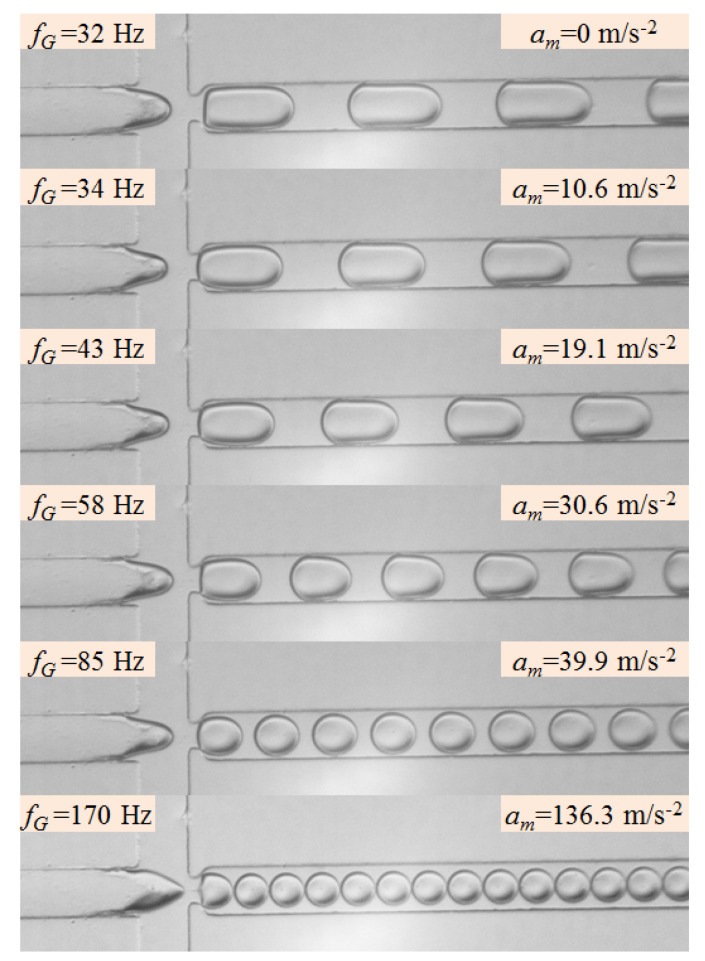
Droplet generation snapshots at different vibration amplitudes. (*Q_c_* = 60 μL/h, *Q_d_* = 60 μL/h, *f_P_* = 170 Hz; *f_G_* are marked on the upper left corner; *a_m_* are marked on the upper right corner.).

**Figure 7 micromachines-11-00743-f007:**
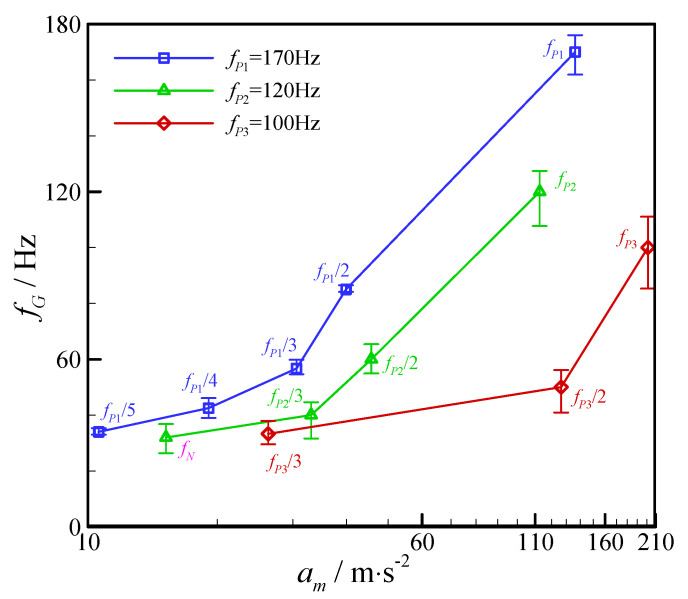
Variation of droplet generation frequency with external vibration amplitude in the flow-focusing microfluidic device. (*Q_c_* = 60 μL/h and *Q_d_* = 60 μL/h.).

**Figure 8 micromachines-11-00743-f008:**
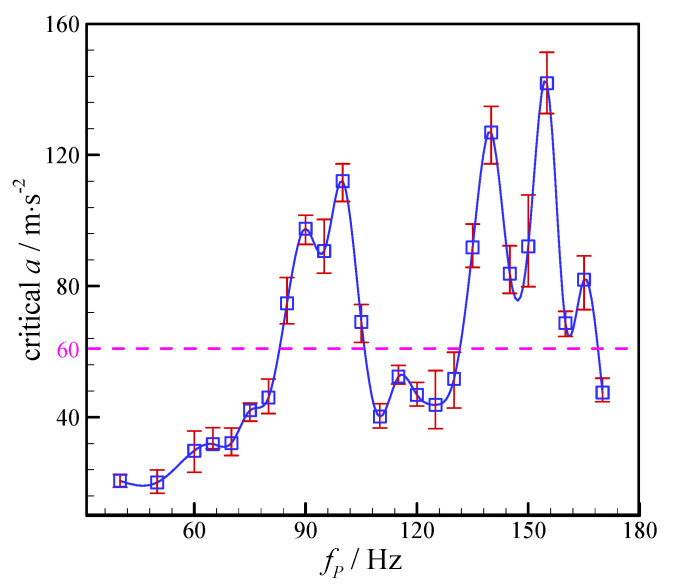
Variation of critical amplitude with external vibration frequency. (*Q_c_* = 90 μL/h and *Q_d_* = 60 μL/h.).

**Figure 9 micromachines-11-00743-f009:**
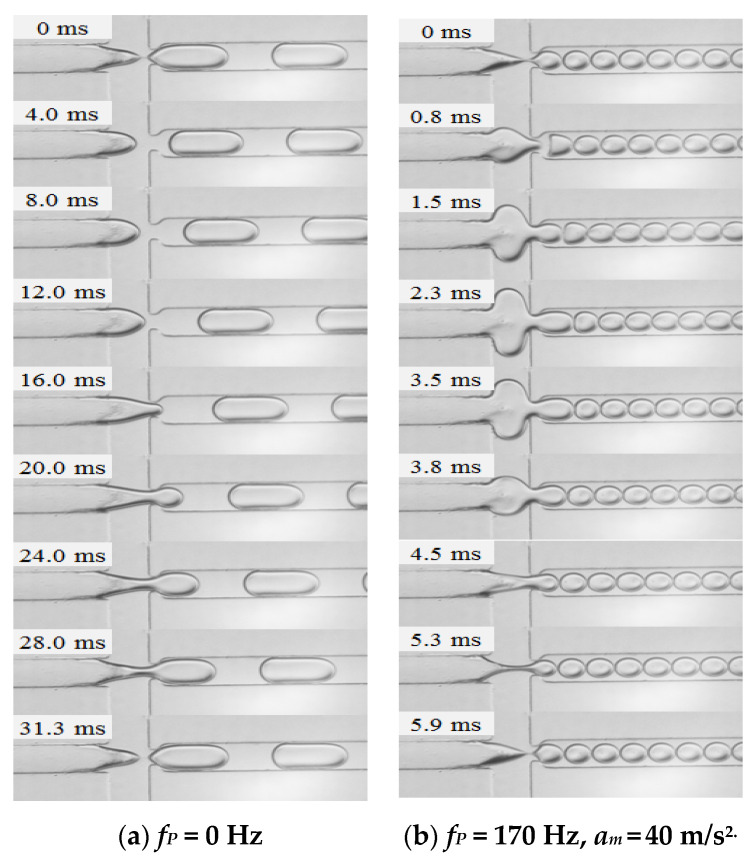
Evolutions of droplet generation process without/with mechanical vibration in the flow-focusing microfluidic device. (*Q_c_* = 60 μL/h and *Q_d_* = 60 μL/h.).

**Figure 10 micromachines-11-00743-f010:**
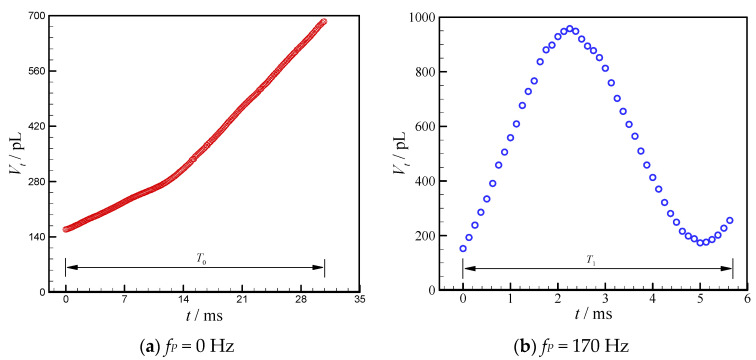
Evolution of dispersed thread head volumes with time during droplet generation without/with external mechanical vibration. (*Q_c_* = 60 μL/h and *Q_d_* = 60 μL/h.).

**Figure 11 micromachines-11-00743-f011:**
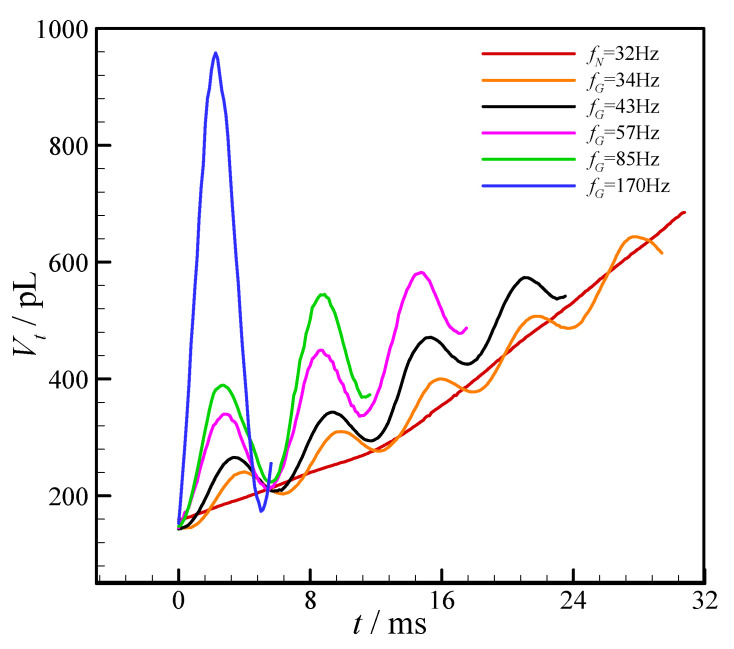
Evolutions of dispersed thread head volume with time during droplet generation process with external mechanical vibration. (*Q_c_* = 60 μL/h, *Q_d_* = 60 μL/h, *f_P_* = 170 Hz, and *a_m_* = 0–136.3 m/s^2^.).

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
