# Peer review of "Droplet Generation in a Flow-Focusing Microfluidic Device with External Mechanical Vibration"

_micromachines, 2020, doi:10.3390/mi11080743_

Round 1

Reviewer 1 Report

In the manuscript titled “Droplet generation in flow-focusing microfluidic device with external mechanical vibration” by Yin et al. focused on the understanding of the influence of mechanical vibration on droplet generation in microfluidic devices. There are quite a few items that need attention before publication.

  1. The abstract needs improvement. There is no basic introduction or more detailed background that is understandable to a scientific investigator in any discipline. Also missing is the general context and broader perspectives that would excite a person to read the paper.
  2. The study lacks novelty. A similar study was published by Zhu et al., Microfluidics and Nanofluidics, 2016. There are quite a few papers that focused on external mechanical vibrations for droplet generations. The authors need to compare and contrast existing literature that is relevant to the manuscript.

It might be a good idea to consolidate all the figures into a smaller number of figures.

Reviewer 2 Report

The work by Yin et al. investigated the effects of frequency and amplitude of external vibration on the droplet generation frequency. They found a critical amplitude beyond which the generation frequency would be synchronized with the vibration frequency. This finding seems important. However, the authors did not find a clear relation of this critical amplitude with frequency or other parameters, which compromises the importance of this findings. Overall, the experiments were well executed. The reviewer recommends its publication upon addressing the following concerns.

  1. The critical amplitude can be useful. It would be helpful to show what parameters contribute to this critical amplitude. For example, can channel geometry, flow rate ratio or even surfactant also affect this critical amplitude? A more comprehensive investigation may be necessary to show the dependence of this amplitude on other factors, which would be beneficial for others when designing such devices.
  2. Introduction needs be better organized. Particularly, discuss more about why the external vibration is advantageous than other methods. Also discuss in detail why the work proposed by the authors is important?
  3. 1, please indicate to which tube the mechanical vibrator is attached and the position of the tube (e.g., is the vibration device attached to the middle point of the tube, the end near the inlet or the beginning near the pump?).
  4. Line 164, the presence of k=1/2 and 1/5 is attributed to geometry and vibration conditions. Please discuss in detail the difference of these two between this work and previous work. Which one (geometry or vibration conditions) is dominant in generating k=1/2 and 1/5?
  5. Fig.4 only shows the relation of fG and fp for one set of flow rate ratio between disperse and continuous phases. Can authors show similar relations for other flow rate ratios?
  6. Authors probably need permission for Fig. 5. Please check.
  7. Why change flow rates for the continuous and disperse phases when investigating amplitude effect? And why 170Hz was picked? In Fig. 8, the flow rates went back to 90 and 60 uL/h, please explain.
  8. Line 310, “1/6 or even smaller of the external vibration frequency droplet generation frequency can be observed”. This assumption lacks supporting data and it seems different from their earlier explanation that channel geometry and vibration conditions contribute together. Please clarify.

Round 2

Reviewer 1 Report

In this revised manuscript, titled “Droplet generation in flow-focusing microfluidic device with external mechanical vibration” by Yin et al. focuses on microfluidic device based droplet generation for various applications. Controlling droplet size was achieved as a function of external mechanical vibration frequency and amplitude. The manuscript has improved considerably. Therefore, I recommend it for publication.

Reviewer 2 Report

Proofread is required before publication.